# Zinc Oxide Zinc Sulfate versus Zinc Oxide Eugenol as Pulp Chamber Filling Materials in Primary Molar Pulpotomies

**DOI:** 10.3390/children8090776

**Published:** 2021-09-02

**Authors:** Moti Moskovitz, Nili Tickotsky, Maayan Dassa, Avia Fux-Noy, Aviv Shmueli, Elinor Halperson, Diana Ram

**Affiliations:** 1Department of Pediatric Dentistry, Hadassah School of Dental Medicine, Hebrew University, Jerusalem 9112102, Israel; motim@md.huji.ac.il (M.M.); fuxavia@gmail.com (A.F.-N.); aviv.dentist@gmail.com (A.S.); elinorhal@gmail.com (E.H.); 2Department of Immunology, Weizmann Institute of Science, Rehovot 76100, Israel; nilitiko@gmail.com; 3Hadassah School of Dental Medicine, Hebrew University, Jerusalem 9112102, Israel; maayan.dassa@mail.huji.ac.il

**Keywords:** pulpotomy, primary molars, Coltosol^®^, intermediate restorative material (IRM)

## Abstract

The long-term effect of Zinc oxide zinc sulfate (Coltosol^®^) dressing material on pulpotomy success and tooth survival has not yet been studied. This study compared the success rates of Zinc oxide zinc sulfate and zinc oxide eugenol as coronal dressing materials post radicular pulp amputation in primary teeth pulpotomies. This study included healthy two- to ten-year-old children who had pulpotomies on primary molars between 2012 and 2018 at the Pediatric Dentistry Clinic of the School of Dental Medicine. Data were analyzed at several follow-ups of up to 60 months. Kaplan-Meier survival curves were used to estimate survival probabilities of Zinc oxide zinc sulfate versus zinc oxide eugenol. In the 107 children included in this study, 54 teeth were filled with Zinc oxide zinc sulfate and 53 were filled with zinc oxide eugenol. Follow-up ranged from 12.2 to 73.3 months. Overall survival of Coltosol^®^ vs. IRM filled teeth was 87.1% and 79.3%, respectively. Overall survival probabilities for Coltosol^®^-filled teeth at 15.5, 24 and 45 months were 95%, 89.8% and 79.7%, respectively, while for IRM they were 93.7%, 83% and 67.7%, respectively. Treatment failure rates and type of treated teeth did not differ between boys and girls (*p*-value = 0.77 and 0.87, respectively). Zinc oxide zinc sulfate and zinc oxide eugenol exhibited comparable high long-term success rates of up to five years (*p* = 0.16).

## 1. Introduction

The main objective of pulp therapy in primary teeth is to maintain the integrity and condition of the teeth and their supporting tissues [1]. When caries removal results in a pulp exposure in a tooth with a normal pulp or reversible pulpitis, a pulpotomy is performed providing there is no evidence of radicular pathology. When the coronal tissue is removed, it is mandatory to ensure that the remaining radicular tissue is vital without suppuration, purulence, necrosis, or excessive hemorrhage [2]. The coronal pulp is amputated up to the orifices of the root canals and the remaining vital radicular pulp tissue surface is treated with medicaments such as formocresol, ferric sulfate, sodium hypochlorite, or MTA. Currently, only MTA and formocresol are recommended as the medicament of choice for teeth expected to be retained for at least 24 months [3], due to the sound evidence regarding their biocompatibility. The issue of placing an adequate sealer in the pulp chamber is of critical importance; an ideal endodontic sealer provides a complete microscopic seal and possesses antimicrobial activity without causing an inflammatory response or cytotoxicity [4], and the radicular pulp should remain asymptomatic without adverse clinical signs or symptoms such as sensitivity, pain, or swelling. There should be no postoperative radiographic evidence of pathologic external root resorption, yet internal root resorption may be self-limiting and stable [1]. Commonly used materials for pulp chamber filling are zinc oxide eugenol [5] and an intermediate restorative material (IRM, Dentsply Caulk Milford, DE, USA) which is a polymer-reinforced zinc oxide-eugenol (composition is detailed in Table 1). According to the manufacture the eugenol in the IRM provides a sedative-like influence on hypersensitive tooth pulp, but the material is contraindicated for direct application on dental pulp tissue, i.e., direct pulp capping. The coronal pulp chamber is filled with a suitable base such as zinc oxide eugenol and the tooth is preferably restored with a stainless-steel crown [1,3].

Zinc oxide—eugenol cements are generally considered to produce a mild pulpal response (without inflammation) when placed in the deep cavity and a persistent chronic inflammation with lack of calcific repair has been widely reported when these materials were placed directly on to the exposed pulp. The toxicity of zinc oxide-eugenol cements has been generally attributed to the eugenol component [6]; Very low concentrations of eugenol produce high toxicity in human dental pulp fibroblasts [7]. Zinc oxide is a well-documented antimicrobial material that forms a reactive oxygen and interferes with bacterial membrane proteins [8]. Zinc oxide-eugenol (ZOE) was among the first agents used for pulp tissue preservation, a concept that is preferred over devitalization (mummification) in primary teeth pulpotomies [9]. Studies reporting negative aspects of ZOE pulpotomies revealed that eugenol possesses destructive properties and cannot be placed directly on the pulp [9]; When ZOE is placed in contact with soft tissue, release of eugenol in high concentrations can cause local cell death [10]. ZOE sealers remain popular because of their slow set, low cost, antibacterial properties, and ease of use [11].

Eugenol is known to be a cytotoxic agent that affects a cell’s membrane and respiratory functions. As a result, non-eugenol zinc oxide sealers were developed to avoid issues with post-operative healing. An alternative to ZOE which does not contain eugenol is Coltosol^®^ (Coltène/Whaledent AG, Feldwiesenstrasse 20, 9450, ltstätten/Switzerland), a radio-opaque, white material for temporary filling that is self-setting and hardens by water absorption. Coltosol^®^ is single-component cement designed for short-term temporary applications (its composition is detailed in Table 1). The cavity should be kept moist before application because dehydration of vital teeth cavities might cause short-term pain [12]. Coltosol^®^ inhibits microbial growth when it is in direct contact with saliva. However, this inhibitory effect tends to decrease over time [13].

The long-term influence of this dressing material on the pulpotomy success has not been investigated. The aim of the present study was to compare the efficacy of Zinc oxide zinc sulfate versus zinc oxide eugenol as a coronal pulp chamber filling material after formocresol has been applied to the radicular pulp tissue surface.

Studies published up to date have concentrated on success rate of different materials used on the amputated pulp in the root canal after vital pulp amputation. The time span in most of these studies is 12 to 24 months. We suggest that because the expected survival of the pulpotomized teeth that is up to 10 years one must look at survival time of the tooth itself post pulp amputation. 

We hypothesized that the survival time may be dependent on the biocompatibility of the material placed in the chamber of the amputated pulp. The present study compared the survival rate of pulpotomized teeth that had a biocompatible material placed in the chamber of the amputated pulp, Zinc oxide zinc sulfate, with the survival rate of teeth that had the highly used zinc oxide eugenol placed in the chamber. Zinc oxide zinc sulfate does not contain eugenol and is therefore more biocompatible than zinc oxide eugenol. The study aim to establish a longer survival of Coltosol^®^ which has a clinical importance for treating teeth at a young age that are expected to survive longer.

## 2. Materials and Methods

*Study design.* In this retrospective study, primary molars of healthy two- to 10-year-old children were treated with pulpotomies at their scheduled regular dental treatment at the Pediatric Dentistry Clinic of the School of Dental Medicine, between 2012 and 2018.

*Sample size and power calculation.* We compared the success percentages of the two treatment groups with that of using IRM after the formocresol (91.4 percent) reported in the literature [14]. sample size was calculated for binary outcome equivalence trial with the aforementioned parameters using the Sealed Envelope method (Sealed Envelope Ltd., London, UK) [15]; if there is truly no difference between the IRM and Coltosol^®^ dressing (91.4% in both groups), then 106 patients (53 per group) are required to be 90% certain that the limits of a two-sided 90% confidence interval will reject the hypothesis that a difference exists between the IRM and Coltosol^®^ group of more than 18%. 

*Ethical considerations.* The study protocol was approved by the Institutional Human Subjects Ethics Committee (0450-18-HMO). All procedures performed were in accordance with the ethical standards of the institutional and national research committee. Participating patients were offered no compensation. 

*Data collection.* Data were collected from treatment files completed between 2012 and 2018. Missing data (i.e., failure to show up for a follow-up appointment) was considered a termination of the follow-up for that participating individual.

*Study population and inclusion criteria.* We included in the study primary molars treated with pulpotomy if they fulfilled all following requirements: primary molars with a deep carious lesion but no spontaneous pain; vital pulp exposed by caries; no pulp degeneration shown by either clinical or radiographic evidence i.e., excessive bleeding from the root canal, internal/external root resorption, inter-radicular and/or periapical bone destruction, a swelling or a sinus tract; stainless steel crown restoration of the teeth. A minimal follow-up period of twelve months was required. For each patient, only a single tooth was included in the study. If the patient had more than one tooth treated, one was randomly (by a toss of a coin) picked for the study.

The follow-up term was measured as the time from the completion of pulpotomy treatment to one of the following: pulpotomy failure identification; natural exfoliation; or patient’s last visit for recall checkup. At follow-up appointments the treated teeth were reviewed by one of two experienced pediatric dentists who were investigators in the study (DR or MM). Failure of treatment was affirmed in one or more of the following: internal/pathological external root resorption; furcation radiolucency or a widened periodontal ligament (PDL) (even asymptomatic); tooth extraction resulting from pathology (including unresolved pain); or if a pulpectomy had to be performed. If none of the above was found, the treatment was considered to be successful. We considered internal resorption a failure even with no accompanying clinical symptoms because it indicates that the treatment did not end in a healthy state of the pulpal tissue [16].

A tooth that exfoliated after more than twelve months (with no pathological external resorption) was considered a success. A tooth that failed at any time period was considered a failure at all later time periods. A tooth not available because the patient had failed to show to follow up, was removed from the success rate determination and from the numerator or denominator, in accordance with Coll et al [17].

*Treatment procedure.* Following local anesthetic with 2% Xylocaine Dental with epinephrine 1:100,000 (lidocaine HCl and epinephrine Injection, USP, Dentsply Pharmaceutical, York, PA, USA) and placing a rubber dam, a high-speed 330 SSW diamond bur (SS White Burs, Inc., Lakewood, NJ, USA) with an air-water coolant was used to gain access to the cavity. Dental caries was then removed with low-speed, round steel burs (Emil Lange, Engelskirchen, Germany).

After coronal pulp amputation, hemostasis was attained with a round bur a cotton pellet moistened with non-diluted formocresol (DSI Dental Solutions, Ashdod, Israel) was placed on the cut off pulp for three minutes. Pulp stumps were then covered with either IRM (Dentsply Caulk Milford, DE, USA) or Coltosol^®^ (Coltène/Whaledent AG, Feldwiesenstrasse 20, 9450, ltstätten/Switzerland). All teeth were restored with stainless-steel crowns. Each treatment was completed with a stainless-steel crown in the same visit.

*Statistics.* The data were analyzed at various follow-up periods. The inter-group differences were stratified according to age, gender, and tooth type and were statistically analyzed using Fisher’s exact test and Barnard’s exact test. We applied Kaplan-Meier survival curves to estimate survival probabilities of IRM compared to Zinc oxide zinc sulfate over time. For the Kaplan-Meier and Cox model procedures, the time to treatment failure was defined as time from treatment to time of failure. If the tooth was lost due to exfoliation, the time to failure was considered censored. In addition, a Cox regression model was created with failure as the dependent variable and gender, age, tooth type, and material (IRM/Coltosol) type as independent variables. Statistical analysis was performed with R 3.6.3 software (R Foundation for Statistical Computing, Vienna, Austria) [18]. All tests applied were two-tailed, and a *p*-value of 0.05 or less was considered a statistically significant inter-group difference.

## 3. Results

This study included a total of 107 children, 59 boys (30 in the Coltosol^®^ group and 29 in the IRM group) and 48 girls (24 in the Coltosol^®^ group and 24 in the IRM group). Fifty-four teeth were filled with Coltosol^®^ and 53 were filled with IRM. Follow-up for Coltosol^®^ ranged from 12.2 to 72.1 months (mean = 31.7, median = 25.3), follow-up for IRM ranged from 12.2 to 73.3 months (mean = 30.2, median = 24.9). 

The overall survival rate was 87.1% for Coltosol^®^-filled teeth and 79.3% for IRM filled teeth. Overall survival probability (shown in Figure 1) of Coltosol^®^-filled teeth at 15.5 months was 95% (95 percent confidence interval,) and 93.7% for IRM, at 24 months it was 89.8% for Coltosol^®^ and 83% for IRM and at 45 months it was 79.7% for Coltosol^®^ and 67.7% for IRM.

The earliest radiographic failure of IRM was observed at 13 months, with 60 months being the latest observed failure. The earliest observed radiographic failure of Coltosol^®^ was observed at 15.4 months, with 53.2 months being the latest observed failure. Five Coltosol^®^ and one IRM-filled tooth presented with more than one radiographic finding. The most common radiographic sign was pathological furcation/periapical radiolucency (*n* = 12, IRM = 6, Coltosol^®^ = 6, shown in Table 2). The only variable that significantly influences the risk of treatment failure in this study is the follow up period (Figure 2).

Gender differences: The follow-up period for boys ranged from 12.2 to 73.3 months and for girls it ranged from 12 to 52.5 months. Treatment failure rates and type of treated teeth did not differ between boys and girls (*p*-value = 0.77 and 0.87, respectively). Type of tooth (Table 3) did not affect treatment outcome (*p*-value = 0.85).

## 4. Discussion

The results of the present study show that both Zinc oxide zinc sulfate Coltosol^®^ and zinc oxide eugenol have high long-term success rates of up to five years. Although there were no statistically significant differences between the two materials, the survival curves demonstrated a more favorable success rate of Zinc oxide zinc sulfate over time. 

Studies on pulpotomy in primary molars typically focus on the medicament placed over the amputated pulp in the root canals [18] but rarely investigate the biocompatibility and long-term outcomes of the material used to fill the pulp chamber that is in contact with the pulp tissue for a long time [19,20,21,22]. This study does not aim to evaluate the success rate of pulpotomy using formocresol as a medication on the amputated pulp (this has been done in numerous studies) but looks at the biocompatibility of the material packed on top of that amputated pulp. We show that using a more biocompatible material as suggested in the present study increases the survival of the treated teeth. 

Set ZOE cement consists of zinc oxide particles embedded in a matrix of zinc eugenolate and other components, in modified ZOE preparations such as IRM that give the material most of its strength [10]. Eugenolate encountering free water is hydrolyzed to yield free eugenol and zinc hydroxide. This hydrolysis forms the basis for the bioavailability of eugenol and ultimately determines whether the agent has therapeutic or toxic effects [10], which in turn will determine the probably for long-term success rate of the pulpotomy. IRM was shown [20] to be more cytotoxic than other calcium silicate cements leading to membrane permeability related apoptosis and necrosis of dental pulp stem cells in vitro [19]. Gonzalez-Lara et al. reported a clinical and radiographic success rate of 84.5% obtained with pulpotomy treatment using regular ZOE as the only capping agent applied over the remaining pulp, in a follow-up period of 24 months [23]. Others stated that ZOE was a less-than-ideal material for pulpotomy (with a success rate of about 55%) when compared to other experimental agents [24,25]; Hui-Derksen et al. studied the use of ZOE B&T^®^ (DENTSPLY, Milford, DE, USA) that has the same ingredients as IRM except for the resin components. They examined pulpotomy records from a private clinic where the instruction was to perform pulpectomy or extraction if bleeding of the radicular pulp was not arrested within 60 seconds. They report a 94% success rate in a follow-up period of over four years [26]. This is higher than found in the present study with a success rate of 79.3% after five years, although in the present study formocresol was applied on the pulp stumps *prior* to filling the pulp chamber with either IRM or Coltosol^®^. Rate of success for IRM was like that found in studies on pulpotomy.

We have previously shown, in a bioinformatics analysis of the interactions between eugenol and periapical tissue proteins, that the fact that eugenol activates Prostaglandin-endoperoxide synthase (PTGS2, also known as cyclooxygenase 2-COX2), a protein involved in both proliferation and apoptosis, suggests that it is responsible for the prostaglandin products involved in inflammation and mitogenesis [27]. While low concentrations of eugenols exert anti-inflammatory and local anesthetic effects on the dental pulp tissue, a high concentration of eugenol is cytotoxic [22]. The concentration of eugenol in the IRM is dependent on the mixing procedure, rendering the quality of the mixture inconsistent. As Coltosol^®^ comes as single-component cement; the ratios between the ingredients do not vary.

Pulp treatment for extensive decay in primary teeth is generally successful. Pulpotomy with MTA compared to formocresol significantly reduced both clinical and radiological failures. The evidence suggests that MTA may be the most effective medicament to heal the root pulp after pulpotomy of a deciduous tooth [19]. A plausible explanation could point to the isolation that MTA provides between the vital pulp remaining in the root canals of the primary teeth and the overlaying medicament. Because MTA, the preferred material for pulpotomy, is very expensive and thus is not applicable for many clinicians, a more biocompatible material than ZOE should be made available to the practitioner that will increase the survival of the treated teeth. 

When in contact with ZOE over time, the deleterious effect of the eugenol results in inflammation and failure of the treatment while no such effect exists when using Coltosol^®^. To the best of our knowledge this is the first attempt to look for an association between the biocompatibility of the material used to pack the pulp chamber and the survival of the treated tooth. To verify the importance of our findings, corroboration of more studies is needed.

A limitation of this study is that it was not a randomized control study, and the sample size was calculated for binary outcome equivalence trial. Equivalence outcome trial was chosen because at the time the authors designed this study no data regarding the use of Coltosol^®^ was available. Based on the results of the present study a randomized control study with a larger sample size will be required for detecting a significant difference in the success rate of pulpotomy with formocresol dressed with Coltosol^®^ versus IRM.

## 5. Conclusions

Zinc oxide zinc sulfate shows comparable (i.e., higher, though not statistically significant) success rate to IRM in human primary molars pulpotomies followed for up to five years.

Survival probability of Zinc oxide zinc sulfate was higher in the long run compared to IRM, although the difference was not significant.

Based on this study’s results, more extensive studies should be performed to determine whether the more biocompatible Coltosol^®^ should be preferred over IRM for human primary molar pulpotomies.

## Figures and Tables

**Figure 1 children-08-00776-f001:**
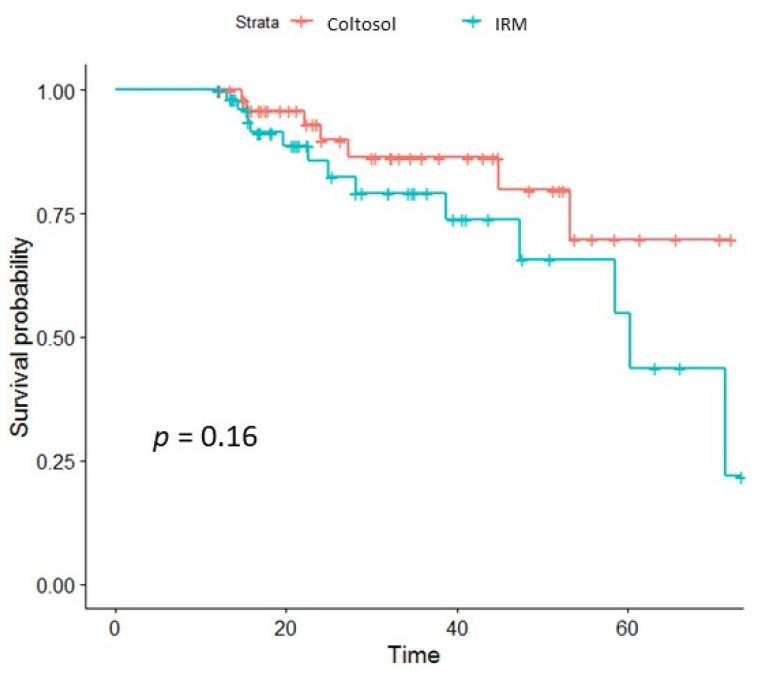
Kaplan-Meier survival curves: survival of Coltosol vs. survival of IRM filled teeth post pulpotomy. Crosshatches indicate censoring time.

**Figure 2 children-08-00776-f002:**
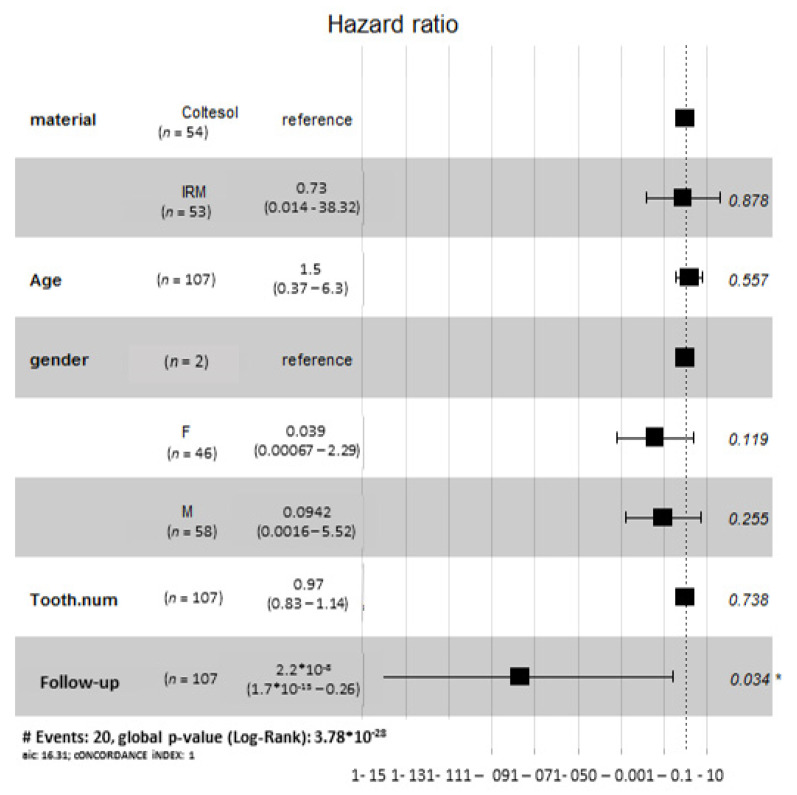
Cox proportional hazards model results. Failure is the dependent variable. The independent variables are gender, age, tooth type, and material (IRM/Coltosol) type. The analysis showed that the only variable that significantly influences the risk of treatment failure in this study is the follow up period.

**Table 1 children-08-00776-t001:** Material ingredients.

Material	IRM (Dentsply Caulk Milford, DE, USA)	Coltosol^®^ (Coltène/Whaledent AG, Switzerland)
Ingredients	Powder: Zinc oxide Poly-Methyl Methacrylate (PMMA) Pigment Liquid: eugenol Acetic acid	Zinc oxide Zinc sulphate-1-hydrate Calcium sulphate-hemihydrate Diatomaceous earth EVA resin Natrium fluoride Peppermint aroma

**Table 2 children-08-00776-t002:** Reasons for treatment failure.

Time (Months)	12–24	24–36	36–48	48–60
IRM	Internal resorption—3	Internal resorption—2		Internal resorption—3
	Pathological radiolucent defect—4		Pathological radiolucent defect—2	
	One patient had both sings *			
Coltosol^®^	Internal resorption—3	Internal resorption—2		
	Pathological radiolucent defect—4	Pathological radiolucent defect—1	Pathological radiolucent defect—1	
	Three patients had both signs *	One patient had both signs *		

* The most common radiographic sign was pathological furcation/periapical radiolucency.

**Table 3 children-08-00776-t003:** Treatment and failures by tooth Type.

Tooth Number	54	55	64	65	74	75	84	85	Total
Coltosol^®^	3	5	4	7	8	13	6	8	54
failure		1			1	3	1	1	7
IRM	2	7	4	7	9	3	11	10	53
failure		3		2	4		2	2	13

FDI World Dental Federation Two-Digit Notation (international) System for Tooth Numbering (ISO-3950).

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
