# Peer review of "Zinc Oxide Zinc Sulfate versus Zinc Oxide Eugenol as Pulp Chamber Filling Materials in Primary Molar Pulpotomies"

_children, 2021, doi:10.3390/children8090776_

Round 1
Reviewer 1 Report
This research is under the scope of this journal; the topic is relevant for readers.
However, there are serious concerns about the present manuscript:
Title - Retrospective study?? But the treatment was select randomly (by a toss of a coin).
Abstract
- It must be reformulated, following the indications for the realization of the abstract, without the words: Objectives, study design, methods and results, and conclusions.
- The use of personal pronouns should be avoided. Example “We compared”.
- In the results, is important to show more information, add some of the p-values.
Introduction
- It should guide the reader to the objective of this work, for this reason, need to be reformulated the introduction.
- (Statement of Relevance)
- What is the importance of this study for clinical? What is the gap in this field of literature?
- You do not think this study is included in the others already done? Which results are comparable with other studies? What has this study been new?
- Please, correct some typos in the all manuscript.
Materials and Methods
- Why you used Materials in a VPT? When the manufacture contraindicated for direct application on dental pulp tissue!! How did the ethical committee approve these procedures?
- How was the sample calculated? Did the authors perform a power analysis to evaluate, if this sample size was appropriate?
- When mentioning materials or devices: please, mention the manufacturer and city/ country.
Discussion
- Please, clarified other limitations of this study?
- And, clarified the future perspectives.
References
The titles of references have a different format, the title of the article is written in capital letters at the beginning of words, others only in lower case. Also, the standardized format of presentation in the journal's name. Because names have been written in a different format, one is not abbreviated, others are not.
Author Response
Reviewer 1
This research is under the scope of this journal; the topic is relevant for readers.
However, there are serious concerns about the present manuscript:
Title - Retrospective study?? But the treatment was select randomly (by a toss of a coin.(
We accept the correction the title was modified.
Abstract
It must be reformulated, following the indications for the realization of the abstract, without the words: Objectives, study design, methods and results, and conclusions.
Abstract was amended as requested.
The use of personal pronouns should be avoided. Example “We compared”.
Personal pronouns were omitted
In the results, is important to show more information, add some of the p-values.
More information was added to the results.
Introduction
It should guide the reader to the objective of this work, for this reason, need to be reformulated the introduction.
(Statement of Relevance)
The following sentences were added to the introduction:
Currently, only MTA and formocresol are recommended as the medicament of choice for teeth expected to be retained for at least 24 months 3, due to the sound evidence regarding their biocompatibility. The issue of placing an adequate sealer in the pulp chamber is of critical importance; An ideal endodontic sealer provides a complete microscopic seal and possesses antimicrobial activity without causing an inflammatory response or cytotoxicity4, and the radicular pulp should remain asymptomatic without adverse clinical signs or symptoms such as sensitivity, pain, or swelling.
…
Vital pulp amputation is a procedure for removing coronal pulp that is inflamed or infected as a result of deep caries, and preserving the root pulp is a vital (living) state. Formocresol is applied to the remaining root pulp to restore it. Currently, it is the most widely used material, which effectiveness has been proven by a large number of research works. Essentially, Formocresol acts through the aldehyde group of formaldehyde. It forms bonds with the side groups of amino acids, both proteins of the bacteria and proteins of the remaining pulp tissue. Therefore, Formocresol is both a bactericidal and a mummifying agent. It kills bacteria and pulp tissue and converts them into inert compounds.
What is the importance of this study for clinical? What is the gap in this field of literature?
We added these paragraphs at the end of the introduction:
"Studies published up to date have concentrated on success rate of different materials used on the amputated pulp in the root canal after vital pulp amputation. The time span in most of these studies is 12 to 24 months. We suggest that because the expected survival of the pulpotomized teeth that is up to 10 years one must look at survival time of the tooth itself post pulp amputation.
We hypothesized that the survival time may be dependent on the biocompatibility of the material placed in the chamber of the amputated pulp. The present study compared the survival rate of pulpotomized teeth that had a biocompatible material placed in the chamber of the amputated pulp, Zinc oxide zinc sulfate, with the survival rate of teeth that had the highly used zinc oxide eugenol placed in the chamber. Zinc oxide zinc sulfate does not contain eugenol and is therefore more biocompatible than zinc oxide eugenol. The study established a longer survival of Coltosol which has a clinical importance for treating teeth at a young age that are expected to survive longer."
You do not think this study is included in the others already done? Which results are comparable with other studies? What has this study been new?
We added the following paragraph in the beginning of the discussion:
This study does not aim to evaluate the success rate of pulpotomy using formocresol as a medication on the amputated pulp (this has been done in numerous studies) but looks at the biocompatibility of the material packed on top of that amputated pulp. We show that using a more biocompatible material as suggested in the present study increases the survival of the treated teeth.
Please, correct some typos in all manuscript.
Were corrected
Materials and Methods
Why have you used Materials in a VPT? When the manufacture contraindicated for direct application on dental pulp tissue!! How did the ethical committee approve these procedures?
In pulpotomy treatments formocresol is applied directly on the amputated pulp after obtaining hemostasis and then covered with ZOE.
According to the manufacturer, the use of formocresol is indicated for treatment of severely infected root canals, effective pulp devitalization and disinfection and root canal dressing. In pediatric dentistry it is used as an additional devitalizing medicine for the secondary treatment on teeth and for root canals. Formocresol is an extremely effective pulp devitalizer, has a pain-relief effect, is a powerful disinfectant and root canal dressing for primary teeth pulp preparation and does not influence physiological root resorption of primary tooth.
According to Cochrane Database of Systematic Reviews by Smaïl‐Faugeron et al. 2018, pulp treatment for extensive decay in primary teeth, all intermediate restorations after the application of formocresol in all the studies consisted of either ZOE or IRM. (Smaïl-Faugeron et al. 2018)
How was the sample calculated? Did the authors perform a power analysis to evaluate if this sample size was appropriate?
This section appears in the manuscript:
"Sample size and power calculation. We compared the success percentages of the two treatment groups with that of using IRM after the formocresol (91.4 percent) reported in the literature14. sample size was calculated for binary outcome equivalence trial with the aforementioned parameters using the Sealed Envelope method (Sealed Envelope Ltd, London, UK)15; if there is truly no difference between the IRM and Coltosol® dressing (91.4% in both groups), then 106 patients (53 per group) are required to be 90% certain that the limits of a two-sided 90% confidence interval will reject the hypothesis that a difference exists between the IRM and Coltosol® group of more than 18%."
When mentioning materials or devices: please, mention the manufacturer and city/ country.
Was added:
formocresol (DSI Dental Solutions, Ashdod, Israel)
Discussion
Please, clarified other limitations of this study?
To the best of our knowledge this is the first attempt to look for an association between the biocompatibility of the material used to pack the pulp chamber and the survival of the treated tooth. To verify the importance of our findings, corroboration of more studies is needed.
And, clarified the future perspectives.
The following paragraph was added to the discussion:
Because MTA, the preferred material for pulpotomy which creates a barrier between the vital pulp and the packed material in the pulp chamber is very expensive and thus is not applicable for many clinicians, a more biocompatible material than ZOE should be made available to the practitioner that will increase the survival of the treated teeth. The present study points to such material.
References
The titles of references have a different format, the title of the article is written in capital letters at the beginning of words, others only in lower case. Also, the standardized format of presentation in the journal's name. Because names have been written in a different format, one is not abbreviated, others are not.
Format was corrected the rest of the formatting was provided by the journal.
Reviewer 2
Comments and Suggestions for Authors
Point 01
“The follow-up period for boys ranged from 12.2 to 73.3 months and for girls it ranged from 12 to 52.5 months (p-value< 0.001).”
Which statistical test was used to compare these values?
This is detailed in the Methods section: " The inter-group differences were stratified according to age, gender, and tooth type and were statistically analyzed using Fisher’s exact test and Barnard's exact test".
Point 02
“Gender differences: The follow-up period for boys ranged from 12.2 to 73.3 months and for girls it ranged from 12 to 52.5 months (p-value< 0.001). Treatment failure rates and type of treated teeth did not differ between boys and girls (p-value = 0.77 and 0.87, respectively). Type of tooth (table 4) did not affect treatment outcome (p-value = 0.85).”
One cannot use tests for analysis of contingency tables to compare failures rates between boys and girls, as the follow-up of the individuals varied. The author will have to use a survival analysis test instead.
We added Cox proportional hazards model results in Figure 2.
The way I see these data, it would be more appropriate to perform a Cox regression having failure as the dependent variable and gender, age, tooth type, and cement type as independent variables.
We added Cox proportional hazards model results in Figure 2.
A table showing the treatment success versus failure for each follow-up period is inadequate (Table 2), as it does not take into consideration the other variable into account.
We believe that Table 2 should be kept, because it serves as a comparison to other studies that use similar tables.
Point 03
Most part of the Discussion consists of a short literature review on the subject without an actual discussion of the findings of the study. The authors need to address this issue.
We added the following paragraph in the beginning of the discussion:
"This study does not aim to evaluate the success rate of pulpotomy using formocresol as a medication on the amputated pulp (this has been done in numerous studies) but looks at the biocompatibility of the material packed on top of that amputated pulp. We show that using a more biocompatible material as suggested in the present study increases the survival of the treated teeth. "
Reviewer 2 Report
Point 01
“The follow-up period for boys ranged from 12.2 to 73.3 months and for girls it ranged from 12 to 52.5 months (p-value< 0.001).”
Which statistical test was used to compared these values?
Point 02
“Gender differences: The follow-up period for boys ranged from 12.2 to 73.3 months and for girls it ranged from 12 to 52.5 months (p-value< 0.001). Treatment failure rates and type of treated teeth did not differ between boys and girls (p-value = 0.77 and 0.87, respectively). Type of tooth (table 4) did not affect treatment outcome (p-value = 0.85).”
One cannot use tests for analysis of contingency tables to compare failures rates between boys and girls, as the follow-up of the individuals varied. The author will have to use a survival analysis test instead.
Actually, the way I see these data, it would be more appropriate to perform a Cox regression having failure as the dependent variable and gender, age, tooth type, and cement type as independent variables. A table showing the treatment success versus failure for each follow-up period is inadequate (Table 2), as it does not take into consideration the other variable into account.
Point 03
Most part of the Discussion consists of a short literature review on the subject without an actually discussion of the findings of the study. The authors need to address this issue.
Author Response
Reviewer 2
Comments and Suggestions for Authors
Point 01
“The follow-up period for boys ranged from 12.2 to 73.3 months and for girls it ranged from 12 to 52.5 months (p-value< 0.001).”
Which statistical test was used to compare these values?
This is detailed in the Methods section: " The inter-group differences were stratified according to age, gender, and tooth type and were statistically analyzed using Fisher’s exact test and Barnard's exact test".
Point 02
“Gender differences: The follow-up period for boys ranged from 12.2 to 73.3 months and for girls it ranged from 12 to 52.5 months (p-value< 0.001). Treatment failure rates and type of treated teeth did not differ between boys and girls (p-value = 0.77 and 0.87, respectively). Type of tooth (table 4) did not affect treatment outcome (p-value = 0.85).”
One cannot use tests for analysis of contingency tables to compare failures rates between boys and girls, as the follow-up of the individuals varied. The author will have to use a survival analysis test instead.
We added Cox proportional hazards model results in Figure 2.
The way I see these data, it would be more appropriate to perform a Cox regression having failure as the dependent variable and gender, age, tooth type, and cement type as independent variables.
We added Cox proportional hazards model results in Figure 2.
A table showing the treatment success versus failure for each follow-up period is inadequate (Table 2), as it does not take into consideration the other variable into account.
We believe that Table 2 should be kept, because it serves as a comparison to other studies that use similar tables.
Point 03
Most part of the Discussion consists of a short literature review on the subject without an actual discussion of the findings of the study. The authors need to address this issue.
We added the following paragraph in the beginning of the discussion:
"This study does not aim to evaluate the success rate of pulpotomy using formocresol as a medication on the amputated pulp (this has been done in numerous studies) but looks at the biocompatibility of the material packed on top of that amputated pulp. We show that using a more biocompatible material as suggested in the present study increases the survival of the treated teeth. "
Round 2
Reviewer 1 Report
This research is under the scope of this journal; the topic is interesting for readers and this research deals with potentially significant knowledge to the field and an open new way for future studies.
The authors improved the quality of the manuscript after the reviewer's indications.
Author Response
Reviewer 1
We thank the reviewer for helping us improve the manuscript.
Reviewer 2 Report
Point 01
Concerning my previous point 01:
“The follow-up period for boys ranged from 12.2 to 73.3 months and for girls it ranged from 12 to 52.5 months (p-value< 0.001).”
Which statistical test was used to compared these values?
The authors replied:
“This is detailed in the Methods section: “The inter-group differences were stratified according to age, gender, and tooth type and were statistically analyzed using Fisher’s exact test and Barnard's exact test".”
This certifies that the authors have no basic knowledge on statistics. One cannot use test for categorical variables to compare mean values of continuous variables. In this case, follow-up period.
Point 02
Table 2 is inadequate because it shows plain percentage of the outcome, when the observations varied greatly in follow-up. Therefore, the percentage does not reflect the real picture, as patients with fillings that were followed up for only 12 months (the minimum, as I can see from the results of the study) were not subjected to clinical elements for the same amount of time as the patients who were followed up for more than 4 years, for example. To keep it because “it serves as a comparison to other studies that use similar tables” is a lousy justification. With this the authors are stating: “A lot of other clinical studies used inadequate statistical tests and reported data in the wrong way. Let’s do the same, for the sake of comparison!” Data can only be shown like this if all the subjects in a study had the exact same follow-up time. This is not the case.
Author Response
Reviewer 2
Point 01
Concerning my previous point 01:
“The follow-up period for boys ranged from 12.2 to 73.3 months and for girls it ranged from 12 to 52.5 months (p-value< 0.001).”
Which statistical test was used to compare these values?
The authors replied:
“This is detailed in the Methods section: “The inter-group differences were stratified according to age, gender, and tooth type and were statistically analyzed using Fisher’s exact test and Barnard's exact test".”
This certifies that the authors have no basic knowledge on statistics. One cannot use test for categorical variables to compare mean values of continuous variables. In this case, follow-up period.
We deleted the "(p-value< 0.001)".
However, the follow up periods didn't differ much, as can be seen from the mean and median values:
"12.2 to 72.1 months (mean= 31.7, median= 25.3), follow-up for IRM ranged from 12.2 to 73.3 months (mean= 30.2, median= 24.9)".
Point 02
Table 2 is inadequate because it shows plain percentage of the outcome, when the observations varied greatly in follow-up. Therefore, the percentage does not reflect the real picture, as patients with fillings that were followed up for only 12 months (the minimum, as I can see from the results of the study) were not subjected to clinical elements for the same amount of time as the patients who were followed up for more than 4 years, for example. To keep it because “it serves as a comparison to other studies that use similar tables” is a lousy justification. With this the authors are stating: “A lot of other clinical studies used inadequate statistical tests and reported data in the wrong way. Let’s do the same, for the sake of comparison!” Data can only be shown like this if all the subjects in a study had the exact same follow-up time. This is not the case.
The table was omitted.
This manuscript is a resubmission of an earlier submission. The following is a list of the peer review reports and author responses from that submission.